# BlueFeather, the singleton that wasn't: Shared gene content analysis supports expansion of *Arthrobacter* phage Cluster FE

**Stephanie Demo**[☯]**, Andrew Kapinos**[☯]**, Aaron Bernardino, Kristina Guardino, Blake Hobbs, Kimberly Hoh, Edward Lee, Iphen Vuong, Krisanavane Reddi, Amanda C. Freise, Jordan Moberg Parker**[ID]*

Department of Microbiology, Immunology, and Molecular Genetics, University of California, Los Angeles, Los Angeles, California, United States of America

☯ These authors contributed equally to this work.
* jmobergparker@ucla.edu

**Data Availability Statement:** All FASTA files are available from the phages.db database and/or GenBank. Accession numbers for the FE cluster

## Abstract

Bacteriophages (phages) exhibit high genetic diversity, and the mosaic nature of the shared genetic pool makes quantifying phage relatedness a shifting target. Early parameters for clustering of related *Mycobacteria* and *Arthrobacter* phage genomes relied on nucleotide identity thresholds but, more recently, clustering of *Gordonia* and *Microbacterium* phages has been performed according to shared gene content. Singleton phages lack the nucleotide identity and/or shared gene content required for clustering newly sequenced genomes with known phages. Whole genome metrics of novel *Arthrobacter* phage BlueFeather, originally designated a putative singleton, showed low nucleotide identity but high amino acid and gene content similarity with *Arthrobacter* phages originally assigned to Clusters FE and FI. Gene content similarity revealed that BlueFeather shared genes with these phages in excess of the parameter for clustering *Gordonia* and *Microbacterium* phages. Single gene analyses revealed evidence of horizontal gene transfer between BlueFeather and phages in unique clusters that infect a variety of bacterial hosts. Our findings highlight the advantage of using shared gene content to study seemingly genetically isolated phages and have resulted in the reclustering of BlueFeather, a putative singleton, as well as former Cluster FI phages, into a newly expanded Cluster FE.

## Introduction

Bacteriophages are ubiquitous biological entities with an estimated $10^{31}$ phage particles on Earth. Assuming an average length of 200 nm, they would extend 200 million light years if stacked head-to-tail [1]. Phages are found in all ecosystems in which bacteria exist and function as drivers of bacterial evolution [2]. They exhibit horizontal gene transfer (HGT) with each other and with bacteria, resulting in the diverse and mosaic nature of phage genomes [3]. Despite their incredible prevalence in the environment, phages remain largely understudied [4].

phages used for most analyses are as follows:
BlueFeather (MT024867), Corgi (MH834607),
Idaho (MK757448), Noely (MH834622), Whytu
(MT024870), and Yavru (MT889364).
Representative phage genome sequences from
each Arthrobacter phage cluster were randomly
selected and accessed from https://phagesdb.org/
for the SplitsTree analysis.

**Funding:** The author(s) received no specific
funding for this work.

**Competing interests:** The authors have declared
that no competing interests exist.

Previous research on mycobacteriophages concluded that phages may exhibit a continuum of diversity, wherein genes are constantly being shuffled amongst the phage population, resulting in shared genes and sequences between different clusters [5]. The immense and ever-expanding diversity of phage genomes has historically been categorized in terms of nucleotide sequence conservation, with a minimum 50% nucleotide identity and 50% span length to at least one phage in a cluster to warrant membership [6, 7]. A mass scale study on *Gordonia* phages also identified a spectrum of genetic diversity, as clusters did not have clear boundaries [8]. Numerous phages lacked the requirement of 50% nucleotide identity but shared many genes, suggesting a relatedness not captured by nucleotide comparisons alone. This relatedness was confirmed with a gene content network phylogeny, and subsequently the cluster assignment parameter for *Gordonia* phages [8], and later for *Microbacterium* phages [9], was adjusted to 35% shared gene content with at least one phage in a cluster. Mycobacteriophages, as well as *Gordonia* and *Microbacterium* phages, exhibited this spectrum; however, the extent of diversity varies depending on the current known phage population, which in turn affects how clustering is carried out. *Arthrobacter* phages were previously found to exchange genes more slowly than *Gordonia* phages, and the 50% nucleotide clustering parameter was considered sufficient at the time [8]. Further studies on *Arthrobacter* phages found these phages to be genetically isolated with highly variable gene content for phages that can infect a range of host species. With this great diversity, nucleotide identity was used to separate *Arthrobacter* phages into 10 distinct clusters and 2 singletons [7], and this parameter has been considered sufficient to categorize the limited number of *Arthrobacter* phages until recently.

Singleton phages can serve as the seeds to start new clusters or be extremely distinct, as they lack the nucleotide identity and/or shared genes required for clustering with known phages. In this study, the genome of novel *Arthrobacter* phage BlueFeather was examined for nucleotide and amino acid identity with other known phages. BlueFeather lacked sufficient nucleotide conservation for clustering according to nucleotide-based parameters, and was thus designated a putative singleton. Phage BlueFeather did, however, have notable amino acid conservation and shared gene content with other *Arthrobacter* phages previously assigned to Clusters FE and FI, suggesting it may not be as isolated as its putative singleton status implied. The outcomes of this research on phage BlueFeather provided evidence for the reclustering of phage BlueFeather, as well as phages formerly assigned to Cluster FI, into a newly expanded Cluster FE.

## Materials and methods

### Sample collection and direct isolation

Soil was collected from Los Angeles, CA in a residential area located at 34.05638889° N, 118.445010000° W. Direct isolation of phages was performed by shaking a soil sample and 2X PYCa broth (Yeast Extract 1 g/L, Peptone 15 g/L, 4.5mM CaCl$_2$, Dextrose 0.1%) in conical tubes at 250 RPM at 25°C for 1.5 hours. After incubation, the solution was filtered through a 0.22 μm syringe and spotted onto *Arthrobacter globiformis* B-2979 (*A. globiformis*). Plaque purifications were performed as described previously and a high titer lysate was filter-sterilized to be used in subsequent characterization experiments [10]. Representative plaques were measured using ImageJ [11] and average plaque diameter was calculated.

### Transmission electron microscopy

Transmission electron microscopy (TEM) was performed on BlueFeather lysate. The sample was placed onto a carbon-coated electron microscope grid and stained with 1% uranyl acetate. Phage particles were visualized using the CM120 Instrument (Philips, Amsterdam,

Netherlands), and micrographs were captured. Phage head and tail lengths were measured using ImageJ [11].

## Genome sequencing and assembly

Viral DNA was isolated with the Wizard® DNA Clean-Up System (cat # A7280, Promega, WI, USA). Sequencing libraries were constructed with the NEBNext® Ultra™ II DNA Library Prep kit (New England Biolabs, MA, USA), and sequenced by Illumina-MiSeq at the Pittsburgh Bacteriophage Institute to an approximate shotgun coverage of 3538x. Genome assembly and finishing were performed as previously described [12].

## Gene annotation

Genomes were annotated as described previously [13] using DNA Master (http://cobamide2.bio.pitt.edu/) and PECAAN (https://pecaan.kbrinsgd.org/) for auto-annotation. GLIMMER [14] and GeneMark [15] were used to predict protein-coding regions along with their start and stop sites. Manual annotation was performed using Phamerator [16], Starterator [17], and host-trained and self-trained GeneMark coding potential maps to support or refute auto-annotation predictions [15]. Gene functions were determined using PhagesDB BLAST (https://phagesdb.org/blastp/), NCBI BLAST [18], HHpred [19] and CDD [20]. The presence of transmembrane proteins was determined using TMHMM [21] and TOPCONS [22]. The annotated complete genome was deposited to GenBank under the accession number MT024867.

## Gene content comparisons

Phage genomes used in this study are available from phagesdb.org [23]. Gepard was used to perform sequence analysis to identify regions of homology between nucleotide sequences or amino acid sequences of different phages [24]. Concatenated whole genome nucleotide and whole proteome amino acid sequences were used to create dot plots with word sizes of 15 and 5, respectively.

SplitsTree was used to generate a network phylogeny in order to reveal the genetic distance between *Arthrobacter* phages [25]. BlueFeather and up to 10 representative phages from each *Arthrobacter* cluster were selected from the Actino_Draft database (version 366) for comparison.

The gene content calculator on PhagesDB (https://phagesdb.org/genecontent/) was used to calculate Gene Content Similarity (GCS), the percentage of shared genes in phams (groups of genes with related sequences), between BlueFeather, Cluster FE, and former Cluster FI phages [16]. Gene Content Dissimilarity (GCD) and maximum GCD gap (MaxGCDGap) were calculated using scripts described previously [8]. Heatmaps and scatter plots were created using Prism 8.0.0 (GraphPad Software, San Diego, California, USA) and were used for quantitative analysis and visualization of GCS and GCD values.

PhagesDB Pham View was used to gather information about phages with genes in the same phams as BlueFeather's [23]. PECAAN was used to obtain the nucleotide sequences for each BlueFeather gene (https://discover.kbrinsgd.org). The BiologicsCorp online GC content calculator was used for each gene in the genome (https://www.biologicscorp.com/tools/GCContent/).

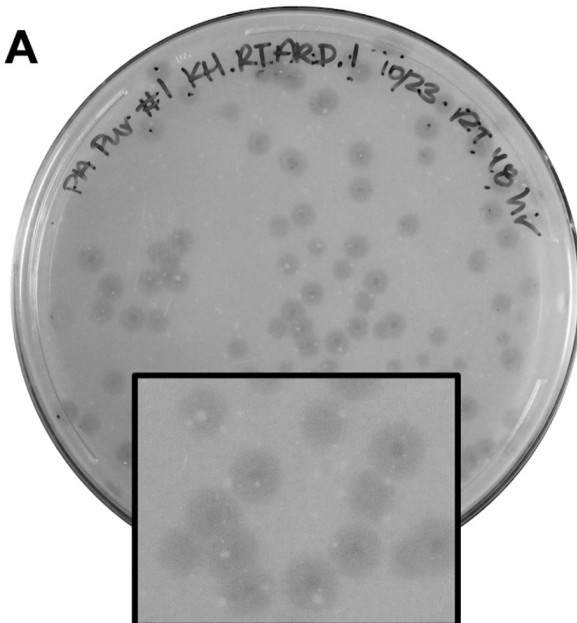
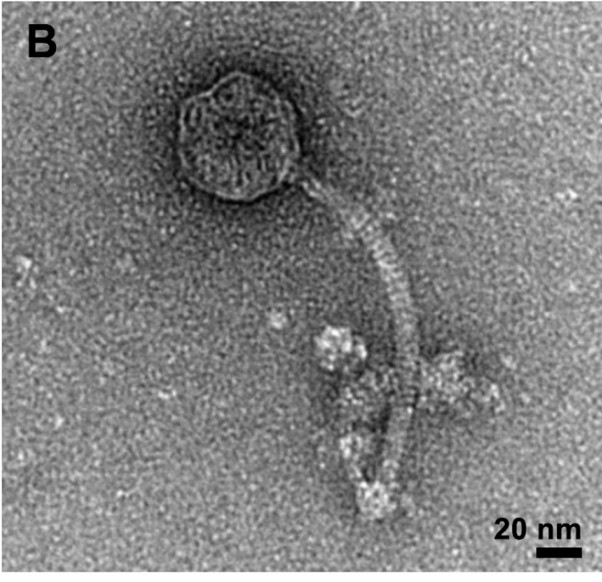

**Fig 1. BlueFeather is a siphovirus. A**. Plaque morphology was inconsistent with some bullseye plaques containing 1 mm center clearings with varying degrees of turbidity. Plaque sizes ranged from 2–5 mm in diameter, with an average plaque diameter of approximately 3.5 mm. **B**. TEM image of BlueFeather at 67,000X magnification. The capsid was estimated to be 48 ± 8 nm and the tail 156 ± 53 nm.

## Results

### BlueFeather is a siphovirus with a short genome

Phage BlueFeather was isolated from a soil sample via direct isolation on *A. globiformis* B-2979 at 25˚C and had a mixed plaque size, ranging from 2–5 mm in diameter (average plaque size of approximately 3.5 mm). Plaque morphology was also inconsistent, with some bullseye plaques containing 1 mm center clearings with varying degrees of turbidity (Fig 1A). Transmission electron microscopy (TEM) at 67,000X magnification showed an average phage capsid diameter and tail length of 48 ± 8 nm and 156 ± 53 nm, respectively (Fig 1B). The long, flexible, non-contractile tail suggested BlueFeather's classification as a *Siphoviridae* [26].

BlueFeather's genome had a length of 16,302 bp, 64.30% GC content, and genome ends with 15 base 3' sticky overhangs (CCACGGTTCCCGTCC). Phages that infect *Arthrobacter* hosts have genome lengths that range from 15,319 bp (Toulouse) to 70,265 bp (PrincessTrina) [7]. The average *Arthobacter* phage genome length (as of May 2020) was 46,968 bp with a standard deviation of 20,619 bp and a median length of 53,859 bp, suggesting that most *Arthrobacter* phages have genomes notably larger than that of BlueFeather. BlueFeather's genome contained 25 manually annotated genes; 18 were of known function, 6 were orphams–meaning they have not been identified in any other known phage–and 1 was a reverse gene (Fig 2). The left arm of the genome had highly conserved genes amongst siphoviral *Arthrobacter* phages, such as those encoding terminase, portal protein, head-to-tail adapter, and tail proteins [7]. Tail tube and sheath genes were absent, confirming the classification of BlueFeather as a siphovirus. Genes characteristic of the lytic life cycle, such as lysin A and holin, were identified; however, there were no genes that would indicate BlueFeather's ability to undergo a lysogenic life cycle, suggesting that BlueFeather is not a temperate phage [27].

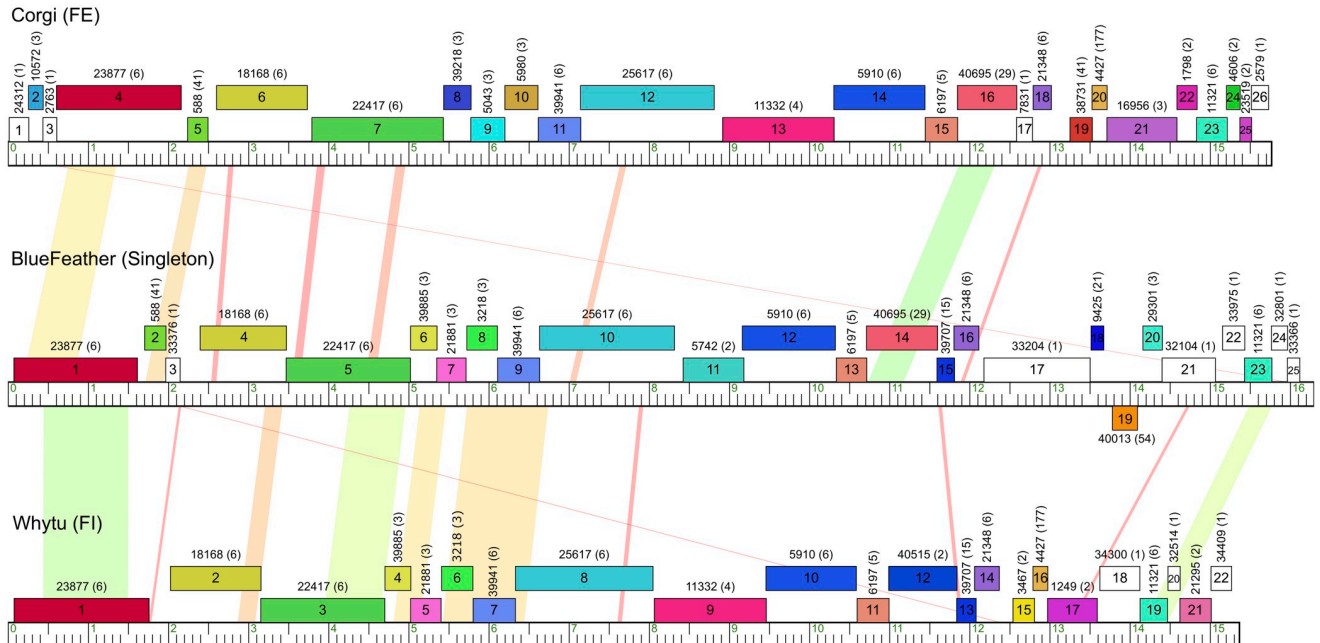

**Fig 2. BlueFeather genome shares little nucleotide similarity but many phams with Cluster FE and former Cluster FI.** The BlueFeather genome is linear with a relatively small length of 16 kbp. Of the 25 identified ORFs, 18 were of known function, 6 were orphams and 1 was a reverse gene. BlueFeather had little BLASTn homology to its most similar phages, as indicated by the limited orange and yellow shading.

## Dot plot comparisons revealed synonymous substitutions in BlueFeather's genome

Phage BlueFeather was originally classified as a singleton on PhagesDB due to low nucleotide identity with other known phages. Nucleotide and amino acid dot plots were created to qualitatively compare BlueFeather to the most similar *Arthrobacter* phages, including those in Cluster FE (Corgi, Idaho, Noely) and former Cluster FI (Whytu, Yavru), as identified by BLASTn. Due to the limited number of sequenced *Arthrobacter* phages, many of the clusters have few members. Of the 28 *Arthrobacter* clusters on PhagesDB (as of May 2020), 17 clusters have between 2–4 phages (including the former Cluster FI). As expected, phages originally assigned to the same cluster had alignments indicating large regions of nucleotide similarity [28], while comparison of BlueFeather's genome to phages originally assigned to Clusters FE and FI revealed no homologous sequences (Fig 3A). Unexpectedly, dot plot analysis of concatenated amino acid sequences with a word size of 5 revealed numerous regions of amino acid sequence similarity between these phages (Fig 3B). This reflects, at present, perhaps one of the clearest examples in which a group of phages lack nucleotide identity while sharing considerable amino acid identity.

## Gene similarity demonstrates a close relationship between BlueFeather and phages of Cluster FE and former Cluster FI

Gene Content Similarity (GCS) is a key metric in quantifying phage genetic relationships and is calculated by averaging the number of shared genes between two phages [29]. GCS was calculated for BlueFeather, Cluster FE phages, and phages originally assigned to Cluster FI. Blue-Feather shared over 35% of genes with all Cluster FE phages, and over 55% of genes with the

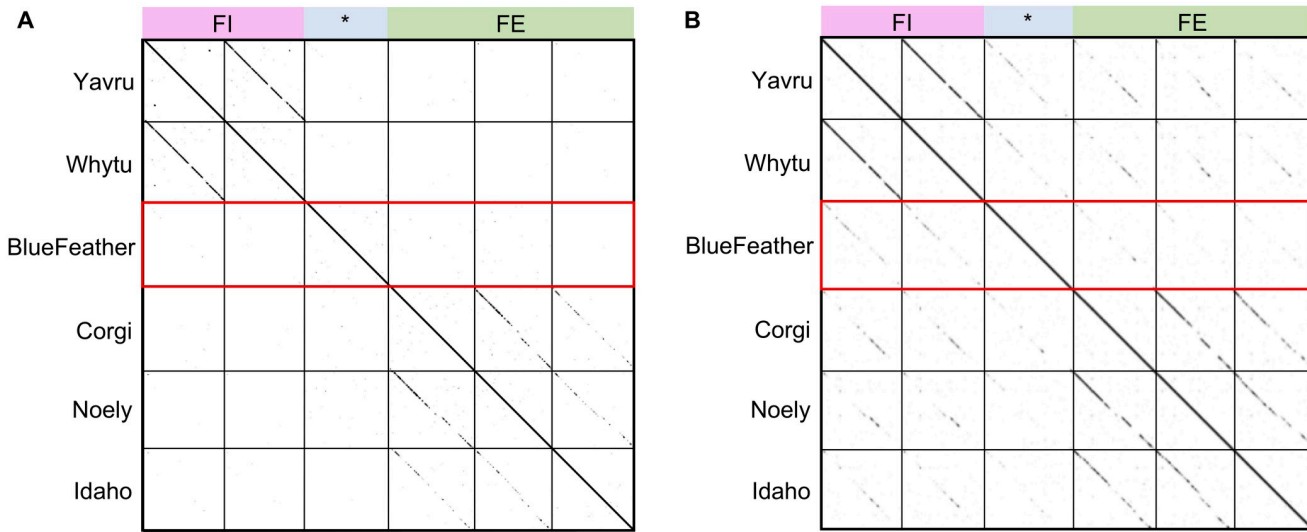

**Fig 3. Dot plots suggest shared amino acids but not nucleotides.** Whole genomes and proteomes for each phage were concatenated and dot plots were created using Gepard. Original cluster information is denoted along the top of each figure, with phage BlueFeather indicated by *. **A**. A whole genome dot plot with word size of 15 indicates strong intracluster nucleotide similarities with both FE and former FI phages. No intercluster nucleotide similarities were observed, indicating BlueFeather does not share significant nucleotide sequences with any of these phages. **B**. A whole proteome dot plot with a word size of 5 indicated the same intracluster amino acid similarities seen in the genome dot plot, but there were also amino acid similarities observed between BlueFeather, Cluster FE, and former Cluster FI phages. BlueFeather appeared to have greater amino acid similarity with phages originally assigned to Cluster FI.

former Cluster FI phages. Over 35% of genes were shared in each pairwise comparison performed (Fig 4A). Given that BlueFeather was originally determined to be a singleton, it was surprising to find GCS greater than the recently adopted threshold of 35% for clustering other phage populations [8, 9]. Gene Content Dissimilarity (GCD) is the opposite of GCS and was used to calculate the maximum GCD gap (MaxGCDGap), a metric that represents the degree of isolation between a phage and a selected phage population [8]. GCD was calculated for Blue-Feather and all *Arthrobacter* phages. There was a MaxGCDGap of 41.60% between BlueFeather

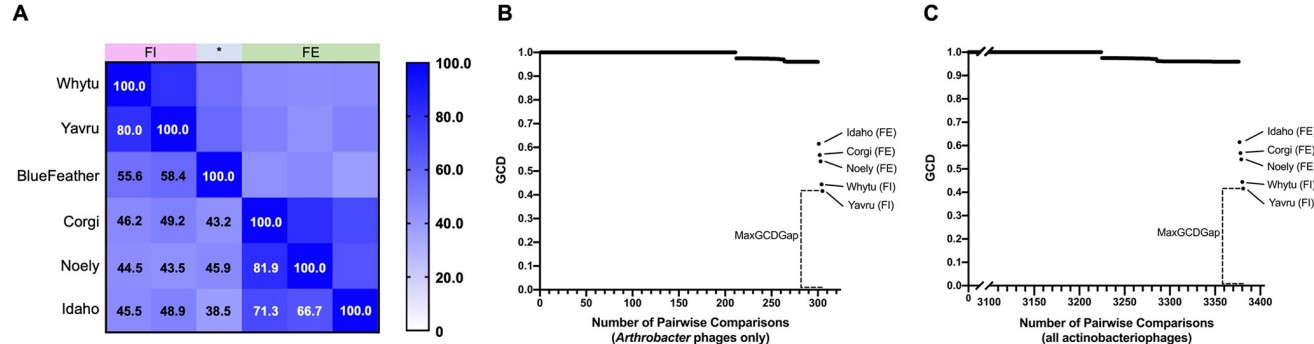

**Fig 4. BlueFeather shares the most phams with phages originally assigned to Cluster FE and former Cluster FI. A**. Gene Content Similarity (GCS) between BlueFeather, Cluster FE and the former Cluster FI was calculated with the PhagesDB GCS calculator using the number of shared phams. There was high intracluster GCS, and BlueFeather showed higher GCS values with former Cluster FI. **B**. Gene Content Dissimilarity (GCD) output values of all pairwise comparisons of BlueFeather and all *Arthrobacter* phages (305), ordered by magnitude. Cluster FE and former Cluster FI were found to be least dissimilar to BlueFeather, with a MaxGCDGap of 41.60%, between BlueFeather and Yavru. **(C)** GCD output values of all pairwise comparisons of BlueFeather and all phages in PhagesDB (3381). MaxGCDGap remained at 41.60%. There are no non-*Arthrobacter* phages that are less dissimilar to BlueFeather than Yavru. BlueFeather shares up to 10% of genes with at least 63 non-*Arthrobacter* phages.

and Yavru, indicating a relatively high degree of separation between BlueFeather and the rest of the *Arthrobacter* phage population (Fig 4B). *Arthrobacter* phages exhibiting pairwise GCD values with BlueFeather of less than 1 were found in Clusters AN, AU, AM, AZ, AV, AL, FE, AO, FH, FF, and former Cluster FI, indicating shared gene content. GCD was then calculated for BlueFeather and all known phages in the PhagesDB Actino_draft database (Fig 4C). Similar to the *Arthrobacter* GCD plot, phages assigned to Cluster FE and former Cluster FI were the least dissimilar to BlueFeather. It is notable that in this comparison, there were 63 additional phages ranging from 0.959 to 0.975 GCD, meaning BlueFeather shares a low number of genes with many non-*Arthrobacter* phages. Non-*Arthrobacter* phages exhibiting pairwise GCD values with BlueFeather of less than 1 were found in *Microbacterium* phage Cluster EE, *Mycobacterium* phage Clusters N, I, P, and the singleton IdentityCrisis, as well *Gordonia* phage Clusters DT, CW and the singleton GMA4.

To compare the relationships between the *Arthrobacter* phage population as whole and the phages assigned to Cluster FE, former Cluster FI, and BlueFeather, a SplitsTree network phylogeny of the phams from each *Arthobacter* phage cluster was generated to examine the genetic distance between the phages. As expected, BlueFeather was shown to be more genetically similar to phages originally assigned to Clusters FE and FI than to any other *Arthrobacter* phage clusters (Fig 5). BlueFeather demonstrated a closer pham similarity to former Cluster FI phages Whytu and Yavru than to Cluster FE phages Idaho, Noely and Corgi; however, these phages altogether formed a distinct branch from the rest of the phages sampled and together comprise the newly expanded Cluster FE.

## BlueFeather genome exhibits evidence of horizontal gene transfer

Given that BlueFeather shares genes with phages infecting distinct hosts, we investigated its genome for potential evidence of horizontal gene transfer (HGT). A whole genome heatmap was created using common metrics for evidence of HGT for each gene in the genome. As of March 2020, 4 genes in BlueFeather were considered to have the most convincing evidence for HGT based on GC content and prevalence in phages that infect unique bacterial hosts: genes 2, 15, 19, and 24 (Fig 6).

Typically, viral genes have about the same [30] or slightly lower GC content [31] compared to their bacterial hosts, suggesting that genes with higher GC content may have been horizontally transferred. BlueFeather had an overall average GC content of 64.30% and *Arthrobacter globiformis* mrc11 was found to have an overall GC content of 65.9% [32]. BlueFeather gene-specific average GC contents ranged from 59.30% to 70.30%, and genes with maximum average GC contents were considered for HGT. This included genes 15 and 24 with GC contents of 70.3% and 70.1%, respectively.

It is increasingly understood that phages infecting different hosts may share considerable gene content through processes such as HGT [8]. For each gene in the BlueFeather genome, we calculated the number of unique isolation hosts for phages possessing a pham found in BlueFeather. Gene 2 belongs to a pham with member genes found in phages that infect *Gordonia malaquae* BEN700 and *Arthrobacter sp*. ATCC 21022. Gene 15 belongs to a pham with member genes found in phages that infect *A. globiformis* B-2979, *A. sp*. ATCC 21022, *Mycobacterium smegmatis* mc[2]155, *G. malaquae* BEN700, and *Gordonia rubripertincta* NRRL B-16540. Gene 19 was the only reverse gene in the BlueFeather genome, and this gene was only found in BlueFeather and in phages infecting *Microbacterium foliorum* NRRL B-24224 SEA and *Microbacterium paraoxydans* NWU1.

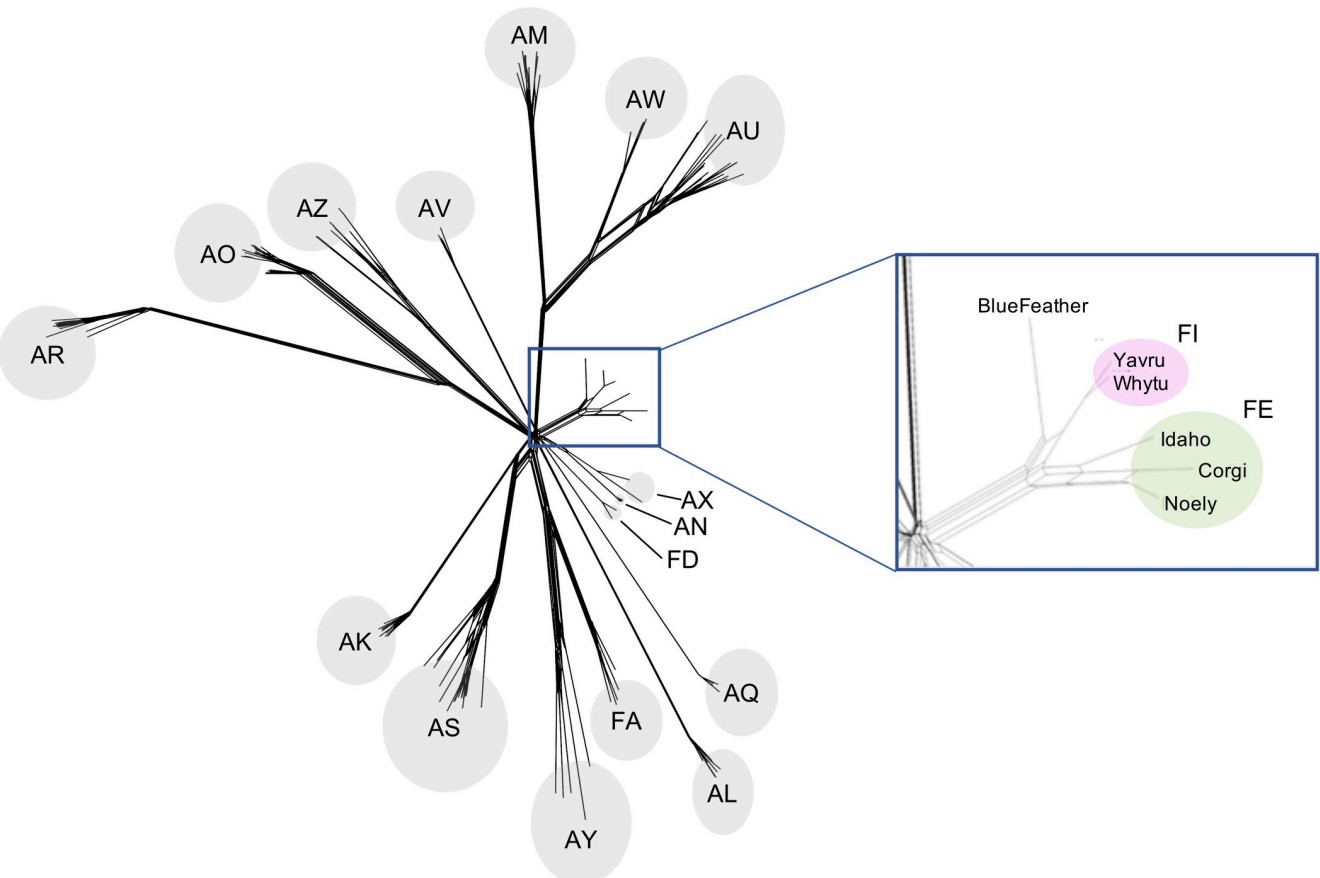

**Fig 5. Expanded Cluster FE includes BlueFeather and former FI phages.** A SplitsTree was generated in order to group *Arthrobacter* phages based on pham similarity. Ten representative phages from each cluster were selected to measure evolutionary relatedness. While there is great diversity of *Arthrobacter* phages, BlueFeather forms a relatively small branch with phages originally assigned to Cluster FE and the former Cluster FI. These phages, boxed in blue, comprise the expanded FE Cluster.

## Discussion

Our research was focused on the genomic and evolutionary relationships between the novel *Arthrobacter* phage BlueFeather and other known phages, particularly those originally assigned to Clusters FE and FI. Previous studies have shown that new clusters can be formed

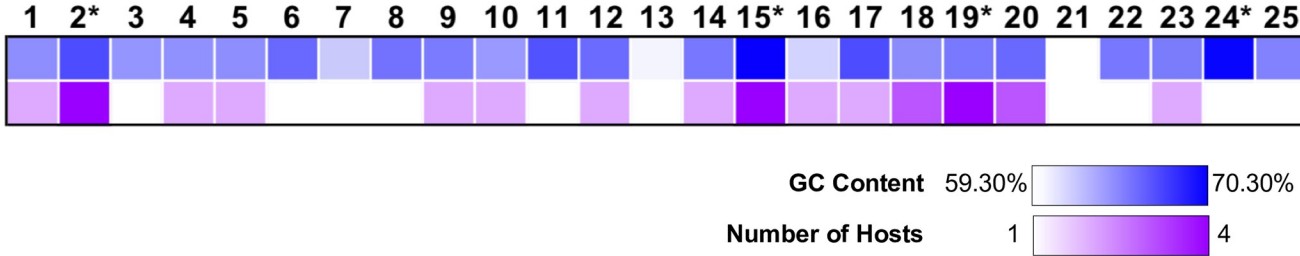

**Fig 6. Evidence of horizontal gene transfer in the BlueFeather genome.** The GC content for each gene in BlueFeather's genome ranged from 59.30%-70.30%, with an average of 64.30%. The number of unique isolation hosts that were represented in each pham ranged from 1–4. Genes with unexpectedly high values were considered to be the result of horizontal gene transfer. There were four genes with the most convincing evidence, indicated by *.

when novel phages are found to be similar to former singletons, as demonstrated by the formation of Cluster AS from *Arthrobacter* singleton Galaxy [7]. BlueFeather, originally designated as a putative singleton phage, exhibits over 35% GCS with all phages originally in Cluster FE and over 55% GCS with those formerly assigned to Cluster FI. The conservation of amino acids, rather than nucleotides, suggests a history of purifying selection via many synonymous mutations in which deleterious mutations were filtered out [33]. Moreover, this presents some of the clearest evidence to date of highly conserved amino acid sequences despite the absence of significant nucleotide conservation amongst related phages.

There is a high degree of synteny between these phages as well. While BlueFeather's original designation as a singleton would imply low genomic relatedness to other phages [5], gene similarity between BlueFeather and phages from Cluster FE and former Cluster FI–in excess of 35%–indicates conservation of gene functions and genome architecture despite extensive divergence of nucleotide identity. While *Arthrobacter* phages have been clustered according to nucleotide identity in the past [7], this study on BlueFeather, Cluster FE, and the former Cluster FI highlights the importance of continually reevaluating clustering parameters, particularly when different parameters may result in different cluster assignments. Moreover, BlueFeather has the smallest genome of all *Arthrobacter* singletons, and it is possible that clustering parameters may also need to take genome size into account. Given that GCS reflects the number of genes shared as a proportion of the total number of genes for each phage, the same number of shared genes would yield higher GCS in comparisons between smaller genomes.

Gene content dissimilarity demonstrated that BlueFeather has a MaxGCDGap of 41.60% with phage Yavru, which was originally assigned to Cluster FI. BlueFeather was found to be least dissimilar with Cluster FE and former Cluster FI phages; this was supported by a network phylogeny of representative *Arthrobacter* phages that indicated great diversity between clusters, but revealed that phage BlueFeather forms a distinct branch with Cluster FE and former Cluster FI phages. Additionally, many phages were found to share between 0–10% GCS with BlueFeather. While this is too low to warrant a significant phylogenetic relationship, it reinforced the observed continuum of diversity in phage populations. Previous research found *Arthrobacter* phage clusters to be very discrete [7]. Even so, this low yet seemingly widespread display of shared genes, as well as BlueFeather's unexpected relationships with Cluster FE and former Cluster FI phages, provides new insight into the genetic landscape of *Arthrobacter* phages. Few phages were previously assigned to Cluster FE and the former Cluster FI, representing only 5 of the 306 sequenced and manually annotated *Arthrobacter* phages (as of May 2020). On the other hand, there are 1,906 sequenced *Mycobacterium* phages (as of May 2020), which has allowed for a more thorough investigation of the mycobacteriophage continuum of diversity. As more *Arthrobacter* phages are sequenced, we expect to observe similar trends in these host-dependent genetic landscapes.

Unlike singleton phages that are replete with orphams [5, 7], the BlueFeather genome, originally designated as a putative singleton, is composed predominantly of genes with known functions that have been assigned to phams. BlueFeather has less than half as many genes as current *Arthrobacter* singletons and contains highly conserved genes required for viral mechanisms. These vital functional genes have been more thoroughly studied and as a result, are more likely to be found in phams with predicted functions [16]. Additionally, given that pham assignments are performed on the basis of amino acid identity, it is unsurprising that many of the phams containing these vital functional genes are shared amongst BlueFeather, Cluster FE, and former Cluster FI phages, despite the lack of significant nucleotide identity in gene encoding sequences.

Markers of horizontal gene transfer (HGT) included unexpectedly high GC content, as well as multiple bacterial hosts on which phages sharing genes with BlueFeather were isolated [5].

BlueFeather shared phams with a multitude of non-*Arthrobacter* phages from various clusters, which allowed us to identify multiple regions as having evidence for HGT. These potential HGT events serve to magnify phage diversity and promote the phenomenon of genetic mosaicism. BlueFeather serves as yet another example of the highly intricate mosaic relationships which exist among phages and are a common feature of the genetic landscape, making phage taxonomy an increasingly difficult task.

In sum, this research has led to the reclustering of BlueFeather and phages formerly assigned to Cluster FI into a newly expanded Cluster FE. Recent observations in which there appear to be limited nucleotide conservation but high shared gene content, as observed in this newly expanded cluster, support the notion that clustering methods should be continually reevaluated and optimized as more phages are sequenced [8]. This study thus provides valuable insight into the continuum of diversity amongst *Arthrobacter* phages, while also supporting a 35% shared gene content clustering parameter as was previously adopted for *Gordonia* and *Microbacterium* phages [8, 9]. Further investigation into novel phages is essential to understand the complex phage landscape. As more *Arthrobacter* phages are discovered, it is likely that we will discover many more phages like BlueFeather which belong to clusters whose close relationships become apparent only through the lens of shared gene content.

## Acknowledgments

We thank Rebecca A. Garlena and Daniel A. Russell at the Pittsburgh Bacteriophage Institute for genome sequence and assembly; and Travis Mavrich, Welkin Pope, Debbie Jacobs-Sera, and Graham Hatfull with the HHMI Science Education Alliance-Phage Hunters Advancing Genomics and Evolutionary Science (SEA-PHAGES) program for programmatic support. The authors acknowledge the use of instruments at the Electron Imaging Center for NanoMachines supported by NIH (1S10RR23057 to ZHZ) and CNSI at UCLA. We would like to thank the Microbiology, Immunology, and Molecular Genetics Department, and the Dean of Life Sciences Division at UCLA for programmatic support.

## Author Contributions

**Conceptualization:** Stephanie Demo, Andrew Kapinos, Amanda C. Freise, Jordan Moberg Parker.

**Data curation:** Amanda C. Freise, Jordan Moberg Parker.

**Formal analysis:** Stephanie Demo, Andrew Kapinos, Aaron Bernardino, Kristina Guardino, Blake Hobbs, Kimberly Hoh, Edward Lee, Iphen Vuong.

**Investigation:** Stephanie Demo, Aaron Bernardino, Kristina Guardino, Blake Hobbs, Kimberly Hoh, Edward Lee, Iphen Vuong.

**Methodology:** Stephanie Demo, Andrew Kapinos, Krisanavane Reddi, Amanda C. Freise.

**Project administration:** Krisanavane Reddi, Amanda C. Freise, Jordan Moberg Parker.

**Resources:** Krisanavane Reddi.

**Supervision:** Amanda C. Freise, Jordan Moberg Parker.

**Validation:** Aaron Bernardino.

**Visualization:** Stephanie Demo, Andrew Kapinos, Kristina Guardino, Blake Hobbs, Kimberly Hoh, Edward Lee, Iphen Vuong.

**Writing – original draft:** Stephanie Demo, Andrew Kapinos, Aaron Bernardino, Kristina Guardino, Blake Hobbs, Kimberly Hoh, Edward Lee, Iphen Vuong.

**Writing – review & editing:** Stephanie Demo, Andrew Kapinos, Amanda C. Freise, Jordan Moberg Parker.

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
