## [Decision Letter · Decision Letter 0]

10 Dec 2020

PONE-D-20-26741

BlueFeather, the singleton that wasn’t: Shared gene content analysis supports expansion of Arthrobacter phage cluster FE

PLOS ONE

Dear Dr. Moberg Parker,

Thank you for submitting your manuscript to PLOS ONE. After careful consideration, we feel that it has merit but does not fully meet PLOS ONE’s publication criteria as it currently stands. Therefore, we invite you to submit a revised version of the manuscript that addresses the points raised during the review process.

Apologies for the long review process. I would be grateful if you could address the comments from both reviewers in the resubmission. Again, sorry for the delay.

We look forward to receiving your revised manuscript.

Kind regards,

Eric Charles Dykeman, Ph.D.

Academic Editor

PLOS ONE

Journal Requirements:

2)  Thank you for stating the following in the Acknowledgments Section of your manuscript:

[This research was funded in part by the Microbiology, Immunology and Molecular Genetics Department, and the Dean of Life Sciences Division at UCLA.]

 [The author(s) received no specific funding for this work.]

Reviewers' comments:

Reviewer's Responses to Questions

**Comments to the Author**

1. Is the manuscript technically sound, and do the data support the conclusions?

Reviewer #1: Yes

Reviewer #2: Yes

2. Has the statistical analysis been performed appropriately and rigorously? 

Reviewer #1: N/A

Reviewer #2: Yes

3. Have the authors made all data underlying the findings in their manuscript fully available?

Reviewer #1: Yes

Reviewer #2: Yes

4. Is the manuscript presented in an intelligible fashion and written in standard English?

Reviewer #1: Yes

Reviewer #2: Yes

5. Review Comments to the Author

Reviewer #1: Dear Editor,

The manuscript entitled “BlueFeather, the singleton that wasn’t: Shared gene content analysis supports expansion of Arthrobacter phage cluster FE” is a very well written paper utilizing current methodologies to characterize the genome of phage BlueFeather and assign it to a cluster. The data are all very solid in support of the authors’ conclusions with regards to the classification of the phage. The in silico analyses seemed to be performed appropriately for what was being investigated. Again, the conclusions are in accordance with the data generated and presented.

In my opinion, however, the conclusion(s) of the paper is not scientifically novel nor important enough to warrant publication in PLOS One. The classification of a single phage to an already existing cluster does not provide sufficient scientific discovery for publication in this journal. A much deeper inquiry into how gene content similarities or dissimilarities would establish new clusters or reorganization of multiple phages would be helpful. Similar papers have classified many phage simultaneously to establish new phage clustering.

Thank you for the opportunity to comment.

Reviewer #2: Review of ‘BlueFeather, the singleton that wasn’t: Shared gene content analysis supports

expansion of Arthrobacter phage cluster FE’

Summary:

Clustering is used to categorize bacteriophages based upon genomic similarity. Previous analyses using nucleotide identity measures designated the novel Arthrobacter phage, BlueFeather, as a singleton indicating that it was not able to be clustered with other phages. This study investigated the use of gene content and amino acid conservation to cluster phage BlueFeather, as its small genome and extensive synteny with clustered phages warranted reevaluation of its designation. Characterization of BlueFeather in terms of plaque and phage morphology is also presented. The authors determined that shared gene content and amino acid conservation between BlueFeather and members of Cluster FI and Cluster FE and phylogenetic analysis using Splitstree warrant reclustering of BlueFeather and the current Cluster FI and Cluster FE phages into one cluster designated Cluster FE.

Due to the mosaic nature of bacteriophage genomes, clustering phages of a particular host can be challenging, especially if there are a limited number of sequenced phage genomes to work with. As the database of phage genomes has grown, the accuracy for clustering has improved, and for Arthrobacter phages, the number of sequenced phages available to evaluate in the initial publication 3 years ago was 46 while today there are 312 Arthrobacter phage genomes sequenced on phagesdb.com. In addition new tools and methods for evaluating genomic relationships have become available besides simply looking at nucleotide identity. Reevaluating cluster assignments is necessary, and in this case merging a singleton, BlueFeather, and Cluster FI phages with the Cluster FE phages is sound.

The authors presentation of their bioinformatic analyses used in this study is clear, and overall the article is well written. This data has not been published elsewhere, and conclusions presented are clear. The conclusions are supported by several different types of bioinformatic analyses making for a strong case for creation of a new Cluster FE and re-clustering of several phages. There are a few minor issues to address such as wording choices and clarification of plaque purifications, but these do not affect the overall conclusions. I recommend this article for publication.

Major issues: There are no major issues to address.

Minor issues: (listed in order of importance)

• Figure 1A: The plaques are difficult to see and are described as being 5mm in diameter though there is a mixture in plaques sizes on this plate. On pg 7, line 153-155, the figure is described as a plate produced from an infection using a picked plaque, so is this from purification steps before the high titer lysate was produced? Perhaps a clearer image of plaques from the lysate would be better and the plaques would be consistent in size. If plaques are variable in size for this phage, that is not uncommon and should be stated.

• Pg 9, line 219: In the text, Idaho has 34.18% shared gene content with BlueFeather while in Figure 4A itself it says that Idaho has 38.5% shared gene content with BlueFeather (though the text in the figure is hard to read in the pdf). These numbers should be consistent. 38.5% GCS would provide an even stronger argument for clustering BlueFeather with Idaho.

• Pg 5, line 96: A reference should be provided for phage purifications or details about the procedure should be included.

• Pg 6, line 126: may I suggest changing ‘whole amino acid sequences’ to ‘whole proteome amino acid sequences’ to clarify that the entire proteome was examined

• Pg 3, line 62: comma needed after phylogeny

• Pg 5, line 103: comma after (Philips, Amsterdam, Netherlands)

• Pg 10, line 228: comma between phages and which

• Pg 10, line 229: comma between 1 and were

• Pg 10, line 235: comma between phages and which

• Pg 10, line 236: comma between 1 and were

• Pg 16, line 373: may I suggest changing this sentence so that it flows better to ‘BlueFeather serves as yet another example of the highly intricate mosaic relationships which exist among phages and are a feature of the genetic landscape, making phage taxonomy an increasingly difficult task.’

6. PLOS authors have the option to publish the peer review history of their article (what does this mean?). If published, this will include your full peer review and any attached files.

Reviewer #1: No

Reviewer #2: No

---

## [Author Response · Author response to Decision Letter 0]

23 Jan 2021

Response to Reviewer 1:

Thank you for the positive feedback regarding the writing, data, and analyses. We respectfully disagree with the reviewer’s suggestion that the conclusion is not scientifically novel or important enough for publication in PLOS ONE. In addition to reclassifying the novel phage BlueFeather from singleton to Cluster FE, this work did also result in the re-organization of the previously distinct Cluster FI into an expanded Cluster FE. Prior to this work, Arthrobacter phages were clustered using nucleotide identity as described by Klyczek et al. (2017), but now the Hatfull lab (which runs phagedb.org) has adopted the parameters we describe for clustering new Arthrobacter phages. We believe our paper has the strong scientific validity, methodology, and ethical standards necessary for publication in PLOS ONE, which doesn’t select articles based on perceived significance.

Response to Reviewer 2 (Minor Issues to Address):

• Figure 1A: The plaques are difficult to see and are described as being 5mm in diameter though there is a mixture in plaques sizes on this plate. On pg 7, line 153-155, the figure is described as a plate produced from an infection using a picked plaque, so is this from purification steps before the high titer lysate was produced? Perhaps a clearer image of plaques from the lysate would be better and the plaques would be consistent in size. If plaques are variable in size for this phage, that is not uncommon and should be stated.

Thank you for the suggestion, we went back and used ImageJ to calculate average plaque diameter (sentence added to methods at lines 99-100). A revised figure 1A has been submitted with an inset showing a magnification of the plaques, and a revised figure legend. The results section has been revised to reflect a more detailed description of plaque size and morphology variations (lines 150-153).

• Pg 9, line 219: In the text, Idaho has 34.18% shared gene content with BlueFeather while in Figure 4A itself it says that Idaho has 38.5% shared gene content with BlueFeather (though the text in the figure is hard to read in the pdf). These numbers should be consistent. 38.5% GCS would provide an even stronger argument for clustering BlueFeather with Idaho.

The GCS calculation was repeated confirming the 38.5% GCS for Phage Idaho (text updated in lines 221-223). The repeated analysis also indicated that the MaxGCDGap was between BlueFeather and Yavru (41.60%), rather than Whytu. The text (line 230), Figure 4b and legend have been revised.

• Pg 5, line 96: A reference should be provided for phage purifications or details about the procedure should be included.

Reference added at line 99.

• Pg 16, line 373: may I suggest changing this sentence so that it flows better to ‘BlueFeather serves as yet another example of the highly intricate mosaic relationships which exist among phages and are a feature of the genetic landscape, making phage taxonomy an increasingly difficult task.’

Thank you for the suggestion, the sentence has been edited.

The following suggested edits have been incorporated:

• Pg 3, line 62: comma needed after phylogeny

• Pg 5, line 103: comma after (Philips, Amsterdam, Netherlands)

• Pg 6, line 126: may I suggest changing ‘whole amino acid sequences’ to ‘whole proteome amino acid sequences’ to clarify that the entire proteome was examined

The suggested edits below change the meaning of the text, so the sentences were changed from “which exhibited” to “exhibiting” to improve clarity (lines 232 and 240).

• Pg 10, line 228: comma between phages and which

• Pg 10, line 229: comma between 1 and were

• Pg 10, line 235: comma between phages and which

• Pg 10, line 236: comma between 1 and were

We would like to thank Reviewer 2 for the detailed feedback, which helped improve the quality of our manuscript.

---

## [Decision Letter · Decision Letter 1]

26 Feb 2021

BlueFeather, the singleton that wasn’t: Shared gene content analysis supports expansion of Arthrobacter phage Cluster FE

PONE-D-20-26741R1

Dear Dr. Moberg Parker,

We’re pleased to inform you that your manuscript has been judged scientifically suitable for publication and will be formally accepted for publication once it meets all outstanding technical requirements.

Kind regards,

Eric Charles Dykeman, Ph.D.

Academic Editor

PLOS ONE

Additional Editor Comments (optional):

Reviewers' comments:

Reviewer's Responses to Questions

**Comments to the Author**

1. If the authors have adequately addressed your comments raised in a previous round of review and you feel that this manuscript is now acceptable for publication, you may indicate that here to bypass the “Comments to the Author” section, enter your conflict of interest statement in the “Confidential to Editor” section, and submit your "Accept" recommendation.

Reviewer #1: All comments have been addressed

Reviewer #2: All comments have been addressed

2. Is the manuscript technically sound, and do the data support the conclusions?

Reviewer #1: Yes

Reviewer #2: Yes

3. Has the statistical analysis been performed appropriately and rigorously? 

Reviewer #1: Yes

Reviewer #2: Yes

4. Have the authors made all data underlying the findings in their manuscript fully available?

Reviewer #1: Yes

Reviewer #2: Yes

5. Is the manuscript presented in an intelligible fashion and written in standard English?

Reviewer #1: Yes

Reviewer #2: Yes

6. Review Comments to the Author

Reviewer #1: Well written paper describing the reclassification of phage based on shared gene content analysis. The data fully support the authors' conclusions and the previous review improved some formatting and descriptive content in the paper.

Minor:

Line 26 delete 'shifting target' for 'challenge' or 'challenging task'.

Line 27 substitute Mycobacterium for Mycobacteria

Line 386 delete 'thus'

Reviewer #2: (No Response)

7. PLOS authors have the option to publish the peer review history of their article (what does this mean?). If published, this will include your full peer review and any attached files.

Reviewer #1: No

Reviewer #2: No

---

## [Editor Report · Acceptance letter]

3 Mar 2021

PONE-D-20-26741R1 

BlueFeather, the singleton that wasn’t: Shared gene content analysis supports expansion of *Arthrobacter* phage Cluster FE 

Dear Dr. Moberg Parker:

I'm pleased to inform you that your manuscript has been deemed suitable for publication in PLOS ONE. Congratulations! Your manuscript is now with our production department. 

Kind regards, 

on behalf of

Dr. Eric Charles Dykeman 

Academic Editor

PLOS ONE